# Lysyl-tRNA synthetase, a target for urgently needed *M. tuberculosis* drugs

Simon R. Green [1], Susan H. Davis[1], Sebastian Damerow[1], Curtis A. Engelhart [2], Michael Mathieson[1], Beatriz Baragaña [1], David A. Robinson [1], Jevgenia Tamjar[1], Alice Dawson [1], Fabio K. Tamaki[1], Kirsteen I. Buchanan[1], John Post[1], Karen Dowers[1], Sharon M. Shepherd[1], Chimed Jansen[1], Fabio Zuccotto[1], Ian H. Gilbert [1], Ola Epemolu[1], Jennifer Riley[1], Laste Stojanovski[1], Maria Osuna-Cabello[1], Esther Pérez-Herrán[3], María José Rebollo[3], Laura Guijarro López[3], Patricia Casado Castro [3], Isabel Camino[3], Heather C. Kim [2], James M. Bean [4], Navid Nahiyaan[2], Kyu Y. Rhee [2], Qinglan Wang[5], Vee Y. Tan[5], Helena I. M. Boshoff [5], Paul J. Converse[6], Si-Yang Li[6], Yong S. Chang[6], Nader Fotouhi[7], Anna M. Upton[7], Eric L. Nuermberger [6], Dirk Schnappinger [2], Kevin D. Read [1], Lourdes Encinas[3], Robert H. Bates[3], Paul G. Wyatt[1] & Laura A. T. Cleghorn [1] ✉

Tuberculosis is a major global cause of both mortality and financial burden mainly in low and middle-income countries. Given the significant and ongoing rise of drug-resistant strains of *Mycobacterium tuberculosis* within the clinical setting, there is an urgent need for the development of new, safe and effective treatments. Here the development of a drug-like series based on a fused dihydropyrrolidino-pyrimidine scaffold is described. The series has been developed against *M. tuberculosis* lysyl-tRNA synthetase (LysRS) and cellular studies support this mechanism of action. DDD02049209, the lead compound, is efficacious in mouse models of acute and chronic tuberculosis and has suitable physicochemical, pharmacokinetic properties and an in vitro safety profile that supports further development. Importantly, preliminary analysis using clinical resistant strains shows no pre-existing clinical resistance towards this scaffold.

Prior to the Covid-19 pandemic, tuberculosis was the world's leading infectious disease killer, resulting in 1.4 million deaths in 2019[1]. Covid-19 is making the prevalence of tuberculosis worse, due to disrupted routine screening programs leading to reduced detection and a resultant impact in increased deaths[2–4]. There is still an urgent need for new agents to treat tuberculosis, to increase the arsenal of weapons available to combat rising clinical drug resistance, and to shorten the length of treatment. Protein synthesis has long been a rich vein for the identification of novel anti-infective agents. Aminoglycosides, targeting bacterial ribosomes, have until recently been a mainstay of multidrug-resistant (MDR) tuberculosis treatment[5]. The oxazolidinone linezolid, which also targets bacterial ribosomes, was approved in a combination treatment against extensively drug-resistant (XDR) tuberculosis[6]. Additionally, GSK3036656, an oxaborole that works by covalently binding to the editing site on leucyl-tRNA synthetase, has entered clinical trials[7,8].

In this report, the identification of a translational inhibitor series is described, targeted against *M. tuberculosi*s lysyl-tRNA synthetase (LysRS). A crystal structure of the protein and the associated structure-based drug discovery (SBDD) program is reported. Mode of action studies indicate that the compounds from the series are killing *M tuberculosis* by targeting LysRS in cells. DDD02049209 the lead

compound from the series has now entered early toxicity studies, towards declaration as a preclinical development candidate.

## Results and discussion

### Identification of a series targeting lysyl-tRNA synthetase

A growth inhibitory phenotypic screen of the Global Health Chemical Diversity Library (~70,000 compounds) was performed against *M. tuberculosis*. One hit compound (**1**) had a modest $IC_{50}$ of ~20 μM in two media that equated to a minimum growth inhibitory concentration (MIC) of 40 μM (Fig. 1). While this level of potency did not warrant follow-up itself, the hit remained of interest because, simultaneously, **1** was identified in a screen against malarial lysyl-tRNA synthetase ($IC_{50}$ 5 μM). This class of enzyme uses ATP, to ligate the amino acid lysine to its uncharged tRNA, making lysyl-tRNA available for use by the protein synthesis machinery. Inhibitors of malarial lysyl-tRNA synthetase, including the natural product cladosporin, are potent inhibitors of *Plasmodium falciparum* growth[9,10]. Genetic analysis has shown that in *M. tuberculosis* the lysyl-tRNA synthetase gene *(lysS)* is essential[11]. Consequently, **1** represented an opportunity to pursue a target-based drug discovery program against *M. tuberculosis* LysRS with a starting point that was already known to have antibacterial activity. When **1** was evaluated against *M. tuberculosis* LysRS it also had modest activity ($IC_{50}$ 43 μM), comparable to its MIC potency.

To assist in the understanding of the structure-activity relationships (SAR) and aid computational design approaches, the first crystal structure of *M. tuberculosis* LysRS was solved (Supplementary Figure 1A) so that the series could be advanced as an SBDD program. *M. tuberculosis* LysRS consists of an N-terminal anticodon binding domain (residues 1–150) with the typical oligonucleotide binding fold, common to lysyl-, asparaginyl- and aspartyl- tRNA synthetases, consisting of a central six-stranded beta-barrel with loops or alpha helices connecting the strands. The C-terminal catalytic domain (residues 151–505) consists of a largely antiparallel beta-sheet flanked by alpha helices, typical of class II aminoacyl-tRNA synthetases.

Initial SAR exploration for the series: exchanging the pyrimidine for a pyridine, expansion of the cyclopentyl to cyclohexyl, and replacement of the H in the 4-position with F (**2**) were all tolerated and improved potency, although with no selectivity over the human ortholog KARS1 (Fig. 1). The observed improvement in potency was enough to allow the first compound/protein structures to be obtained, following soaking of ligands into Apo crystals. Structural characterization of ligands showed them to bind in the adenosine pocket, stacked between the side chains of Phe269 and Arg481 (Fig. 2a). The pyrimidine core formed hydrogen bonds with the main chain of Ser266 as well as water-mediated contacts through the lactam carbonyl group to Asp480 (Fig. 2a). Thus, the binding site for the LysRS inhibitors was different to the previously mentioned oxaboroles, that covalently bind to the LeuRS editing site, which was not unexpected since LysRS does not contain an editing site[7]. As the SAR for the series developed, the pyridine molecules were not as potent as their pyrimidine equivalents; in addition, the pyridines had lower in vivo exposure and bioavailability, as such, further SAR focussed on the pyrimidine core. Alkyl ether substituents at the 4-position resulted in improved potency towards both the enzyme and bacteria, as seen for **3** and **4**. For the 4-alkyl ethers, a second water molecule was present and the side chain of Asp480 was pushed further out of the binding site, allowing an additional interaction with the water network linking to Asp480 (Fig. 2b highlights this for **8**).

The 4-alkyl ether groups also resulted in significantly improved selectivity against KARS1, in particular the 4-ethoxy, which was correspondingly reflected in excellent selectivity at a cellular level (compare **4** and **5** in Fig. 1). From crystal structures, it was clear that the adenosine binding pocket of LysRS was more flexible than KARS1. Residues Asn258 - Pro267 form a loop over the adenosine binding pocket, which significantly changes conformation on ligand binding in LysRS (Supplementary Figure 1B). In the absence of a ligand, the loop is poorly ordered, while on ligand binding, the loop closes over the catalytic site. In KARS1 (PDB code 3bju) an interaction is formed between this loop and the C-terminus of the protein (involving the carbonyl of Leu329 and the amide of Ile564). No equivalent interactions are formed in LysRS (Ser263 and Ile491). This flexibility allows larger groups to bind in the LysRS adenosine pocket, giving compounds selectivity over KARS1. The 2-hydroxyl in the (1 S,2 R)−2-aminocyclohexan-1-ol substituent of **4** did not appear to contribute to a significant improvement in potency. However, the 2-hydroxyl resulted in improved in vitro metabolic

**1**
LysRS $IC_{50}$ = 43 μM
KARS1 $IC_{50}$ >100 μM
MIC = 40 μM
HepG2 >50 μM

**2**
LysRS $IC_{50}$ = 9 μM
KARS1 $IC_{50}$ = 9 μM
MIC = 0.6 μM
HepG2 = 11 μM

**3**
LysRS $IC_{50}$ = 1.6 μM
KARS1 $IC_{50}$ >100 μM
MIC = 0.8 μM
Intra. Mac. $IC_{90}$ = 0.7 μM
HepG2 >100 μM

**4**
LysRS $IC_{50}$ = 0.7 μM
KARS1 $IC_{50}$ = 32 μM
MIC = 0.3 μM
HepG2 = 46 μM

**5**
LysRS $IC_{50}$ = 0.9 μM
KARS1 $IC_{50}$ >100 μM
MIC = 1.0 μM
Intra. Mac. $IC_{90}$ = 0.8 μM
HepG2 >100 μM

**6**
LysRS $IC_{50}$ = 0.5 μM
KARS1 $IC_{50}$ >100 μM
MIC = 0.5 μM
Intra. Mac. $IC_{90}$ = 0.5 μM
HepG2 = 97 μM

**7**
LysRS $IC_{50}$ = 0.6 μM
KARS1 $IC_{50}$ >100 μM
MIC = 0.9 μM
Intra. Mac. $IC_{90}$ = 0.5 μM
HepG2 >100 μM

**8**
LysRS $IC_{50}$ = 0.05 μM
KARS1 $IC_{50}$ >100 μM
MIC = 0.08 μM
Intra. Mac. $IC_{90}$ = 0.07 μM
HepG2 >100 μM

**Fig. 1 | Chemical evolution of DDD2049209 (8) from the initial hit 1.** Potencies are shown against: in vitro enzymes LysRS & KARS1; *M. tuberculosis* phenotypic growth in culture (MIC), *M. tuberculosis* growth within macrophages, and cytotoxicity against human HepG2 cells. Data shown are the geometric mean of at least two independent replicates. Source data are provided as a Source Data file.

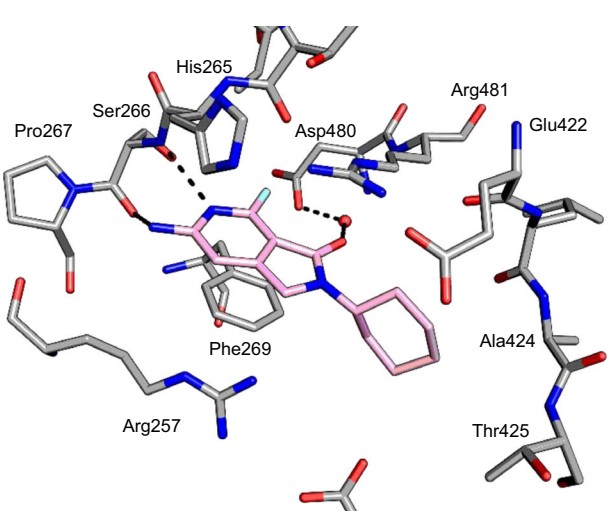

**Fig. 2 | Compounds bound in the LysRS catalytic binding site. a 2** is shown in pink carbon atoms, and surrounding protein residues are shown with gray carbon atoms. Oxygen atoms are shown in red, nitrogen atoms in dark blue, and fluorine in pale blue. Key interactions are indicated with dashes (PDB 7qhn). **b 8** is shown with gold carbon atoms, other atoms as in **a** (PDB 7qi8).

stability, and in vivo exposure was also increased, partially due to significantly better oral bioavailability (compare **3** and **5** in Supplementary Table 1). Though methoxy groups at the 4-position of the pyrimidine were less selective than ethoxy, the addition of a 2,2 difluoro to the cyclohexyl ring resulted in a clear increase in selectivity (**6**). The improved selectivity was proposed to be due to the presence of a more hydrophobic methionine residue in LysRS in place of a hydrophilic threonine in KARS1. Further expansion of the cyclohexyl group to a cycloheptyl also resulted in modestly improved potency and metabolic stability (**7**). The combination of the most promising components of the SAR generated the lead compound DDD02049209 (**8**), with a methoxy in the 4-position, and (1 S,2 S)-2-amino-3,3-difluorocycloheptan-1-ol at R2 (Fig. 1). This molecule had excellent potency $IC_{50}$ 0.05 μM; (95% confidence interval 0.03–0.07 μM, $n = 10$); MIC 0.08 μM (95% confidence interval 0.05–0.1 μM, $n = 5$) and exquisite selectivity (>100-fold) over the human counterpart, at both an enzyme (KARS1 $IC_{50}$ > 100 μM; ($n = 12$) and cellular level (HepG2 $IC_{50}$ > 100 μM; ($n = 6$) (Fig. 1). For the majority of the series, the enzyme $IC_{50}$ and the phenotypic MIC were roughly equivalent. This is not always the case for inhibitors and could be related to a number of factors including the excellent physicochemical properties of the molecules (low molecular weight and high solubility); additionally, high binding affinity and long off-rate may also contribute to the parity in potency. The MIC potency for the series was not impacted when performed in the presence of 1 mM lysine (Supplementary Table 3), unlike mupirocin (with isoleucine)[12]; this was not unexpected, because from the co-crystal structures compound binding did not involve the lysine binding site. In general, the series had good drug-like properties, as exemplified by **5** and **8**, excellent cross-species in vitro metabolic stability, and good in vivo exposure (Supplementary Tables 1 and 2). To progress this compound series toward the clinic, representatives were assessed in a range of standard in vitro safety assessment assays. Both **5** and **8** had a very clean profile when tested in off-target receptor panels, and genotoxicity and cardiotoxicity assays (see methods). Exposure of **8** was also profiled in other preclinical species (rat, dog), displaying good pharmacokinetic properties including excellent oral bioavailability (>70%), an essential prerequisite for use in a first-line tuberculosis treatment regimen (Supplementary Figure 2).

## In vivo assessment

During the evolution of the SAR for the series, compounds with appropriate properties were assessed in pharmacokinetic exposure studies and acute murine in vivo models of infection (Supplementary Table 1). In vivo efficacy, dose-response curves were obtained for the most promising molecules. Both **5** and **8** demonstrated good efficacy, achieving a >3 $log_{10}$ reduction in colony forming units (CFU) with an $ED_{99}$ (dose that causes a 2 $Log_{10}$ reduction in CFU with respect to untreated mice) of 49 and 12 mg/kg respectively (Fig. 3a). While the acute model demonstrated that LysRS inhibitors can kill *M. tuberculosis* in vivo, a patient presenting with TB would already have established chronic disease. As the chronic setting sets a higher challenge for treatment success, only the more potent compound, **8** was evaluated in a chronic model of infection. Compound **8** achieved a clear reduction in lung CFU, with a maximum effect of 2.4 $log_{10}$ reductions and an $ED_{99}$ of 94 mg/kg (Fig. 3b); similar to previously published data of other translation inhibitors in this model[7].

Ultimately, all new tuberculosis drugs will be used in multi-drug combinations. Thus, it was of interest to evaluate the contribution of a LysRS inhibitor to combinations of TB drugs in a preclinical subacute model. An in vivo combination study was performed evaluating the early lead **5** (K) in combination with bedaquiline (B) and pretomanid (Pa), from the recently approved clinical combination BPaL[6]. In this model, K directly replaced the oxazolidinone linezolid (L), as a translation inhibitor with an alternative mode of action (MoA). The dose selected was chosen from an earlier study that indicated 100 mg/kg bid had a bacteriostatic effect in this model. Treatment with Pa resulted in a significant decrease in CFU over the treatment period of 8 weeks. The addition of either B or K contributed to an additional reduction in CFU. The triple combination treatment of BPaK resulted in an even greater reduction in CFU, with lungs from 3 of the 5 mice having no detectable bacteria after two months of treatment (Fig. 3c). Moreover, there was no statistically significant difference between BPaK and BPaL, although in the latter case, all 5 of the mice had no detectable bacteria in their lungs at the end of treatment (Fig. 3c). The data indicate that inhibition of LysRS adds to the bactericidal activity of BPa, two established clinical agents, and LysRS inhibitors can replace linezolid in this preclinical model.

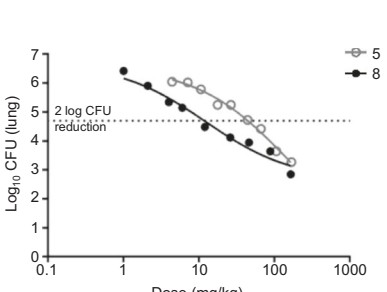
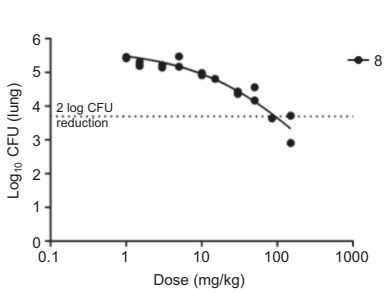
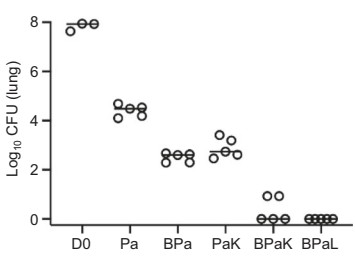

**Fig. 3 | In vivo efficacy of LysRS inhibitors in murine models of *M. tuberculosis* infection. a** Compounds **5** and **8** were tested in an acute model of infection in C57BL/6 mice. Dosing started 1 day after infection and lasted for 8 days; each group consisted of one mouse/dose. The effect on the number of colony-forming units (CFU) in mouse lungs is shown. **b** Compound **8** was tested in a chronic model of infection in C57BL/6 mice. Dosing started 6 weeks after infection and lasted for 8 weeks (7 days/week), each group consisted of two mice/dose. **c** Compound **5** (K: 100 mg/kg bid) was evaluated in a subacute model combination study in BALB/c mice along with bedaquiline (B: 25 mg/kg qd), pretomanid (Pa: 25 mg/kg bid), and linezolid (L: 50 mg/kg bid). Dosing (8 weeks: 5 days/week) started 14 days post infection (D0), five animals/group were used. Source data are provided as a Source Data file.

## Evaluation of cellular MoA

Although the SAR for this series was developed as a LysRS SBDD program, it was important to demonstrate that the compounds were killing *M. tuberculosis* by targeting LysRS in cells. Multiple attempts were made to isolate resistant mutants to **7** as an early representative of the series. At 10 times MIC the frequency of resistance was low ($2.5 \times 10^{-9}$) suggesting a high barrier to resistance. One mutant was isolated after nine weeks growth on media supplemented with **7** (10 μM). This mutant was confirmed to be ~30-fold more resistant to **7** compared to wild-type (WT) H37Rv while displaying equivalent sensitivity to the control drug isoniazid that does not target LysRS (Fig. 4a).

The resistant mutant contained mutations in three genes: *ppsA*, *ppe62*, and *lysS*. The first two genes have been shown to be non-essential[11] and SNPs can be found in these genes when resistant mutants are isolated for a range of unrelated compounds, potentially due to affecting PDIM biosynthesis and PE/PPE-mediated uptake of small molecules[13]. However, the presence of a mutation in *lysS* was a strong indicator of on-target MoA. The mutation corresponded to an in-frame insertion of a Ser residue at position 491, close to the carboxy terminus of the protein, in a region that caps the adenosine binding pocket. The insertion is adjacent to Ile491 (in the WT sequence; Supplementary Figure 1C) which is an invariant residue across species and was the residue for one of the resistant mutations isolated when *Saccharomyces cerevisiae* was treated with cladosporin (I567V)[10].

To explore the impact of the S491 insertion on other inhibitors from the series, the mutated protein was expressed and assessed in vitro. Diverse effects on sensitivity were obtained, primarily related to the size of the substituent at the 4-position that bound in the pocket, close to the site of the S481 insertion, responsible for sensitivity over human KARS1. Thus, when an ethoxy was present, which penetrates into the pocket the furthest and was initially being used to drive selectivity over KARS1 (**5** and **7**), shifts of >35-fold were seen in the IC$_{50}$ against the mutant enzyme compared to the WT (Supplementary Table 4). A 4-methoxy substituent (**8**) that does not enter the pocket as deeply was impacted less by the mutation (4.6-fold). In contrast, a small 4-fluoro-substituent (**2**), which does not enter the selectivity pocket and had no selectivity at the enzyme level, surprisingly showed an increase in inhibition of the mutant protein compared to the WT (Supplementary Table 4). Notably, when the same compounds were tested against the original resistant mutant strain, the effect on MIC correlated with the in vitro enzyme data. Thus, the ethoxy compounds were >30-fold less active against the resistant mutant, likewise, the methoxy was 3.7-fold less active, but in contrast, the 4-fluoro substituent was again more potent against the resistant mutant strain (Supplementary Table 4). To further explore the impact of the S491 insertion on the sensitivity to the series, strains were generated that overexpressed either WT LysRS or the S491 mutant protein, both under the control of the ATc promoter. Overexpression of WT LysRS resulted in an ~30-fold shift in MIC for **8** compared to both H37Rv and the uninduced overexpressing strain (Fig. 4b). Overexpression of the mutant LysRS protein containing the S491 insertion resulted in a >125-fold shift in MIC for **8** (Fig. 4b), thus confirming that the S491 insertion in LysRS was driving the reduced sensitivity to the LysRS inhibitors in the resistant mutant strain. Clearly, mechanisms of resistance other than coding region SNPs are feasible (e.g., upregulation of efflux pumps or *lysS* promoter mutations that influence the expression of LysRS); resistant mutant isolation studies are continuing, now using **8** as the probe molecule.

As a further exploration of the cellular MoA, a metabolomic profile was performed on three inhibitors from the series in comparison to a closely related compound that showed no LysRS inhibition (**9**). A selected metabolite profile showed a clear demarcation between the three active compounds (**5**, **6** & **7**), and the inactive compound (**9**) and DMSO (**D**) control (Fig. 4c). Some of the metabolites on the lysine biosynthetic pathway, including lysine itself, were significantly upregulated following treatment with LysRS inhibitors. In addition, saccharopine, which is part of the lysine degradation pathway in higher eukaryotes (although less clear in Mycobacterium sp.) is significantly downregulated. These data suggest that following treatment with LysRS inhibitors and the resultant stalling of protein synthesis due to limited lysyl-tRNA, the cells act as if starved for lysine and try to compensate by increasing lysine biosynthesis.

## Comparison with oxazolidinones

The oxazolidinone linezolid was recently approved for the treatment of XDR tuberculosis[6]. Oxazolidinones bind to the 50 S ribosomal subunit preventing the formation of the 70 S translational initiation complex[14]. Prolonged clinical treatment with linezolid, as in the approved BPaL regimen, can lead to anemia and other cytopenias, as well as peripheral and optic neuropathy. Each of these toxicities is attributed to linezolid binding to human mitochondrial ribosomes, which is much closer to a prokaryotic ribosome than the cytoplasmic eukaryotic ribosome and thereby inhibiting mitochondrial protein synthesis (MPS)[15,16]. For an antibacterial agent, LysRS is a particularly attractive aminoacyl-tRNA synthetase to pursue, because in eukaryotic cells, unlike the majority of these enzymes, mitochondrial DNA does not encode a specific copy of the enzyme. Instead, the mitochondrial protein derives from differential splicing of the nuclear *KARS1* gene, adding a mitochondrial targeting sequence onto the cytoplasmic protein[17–19]. Thus, the demonstration that LysRS inhibitors did not

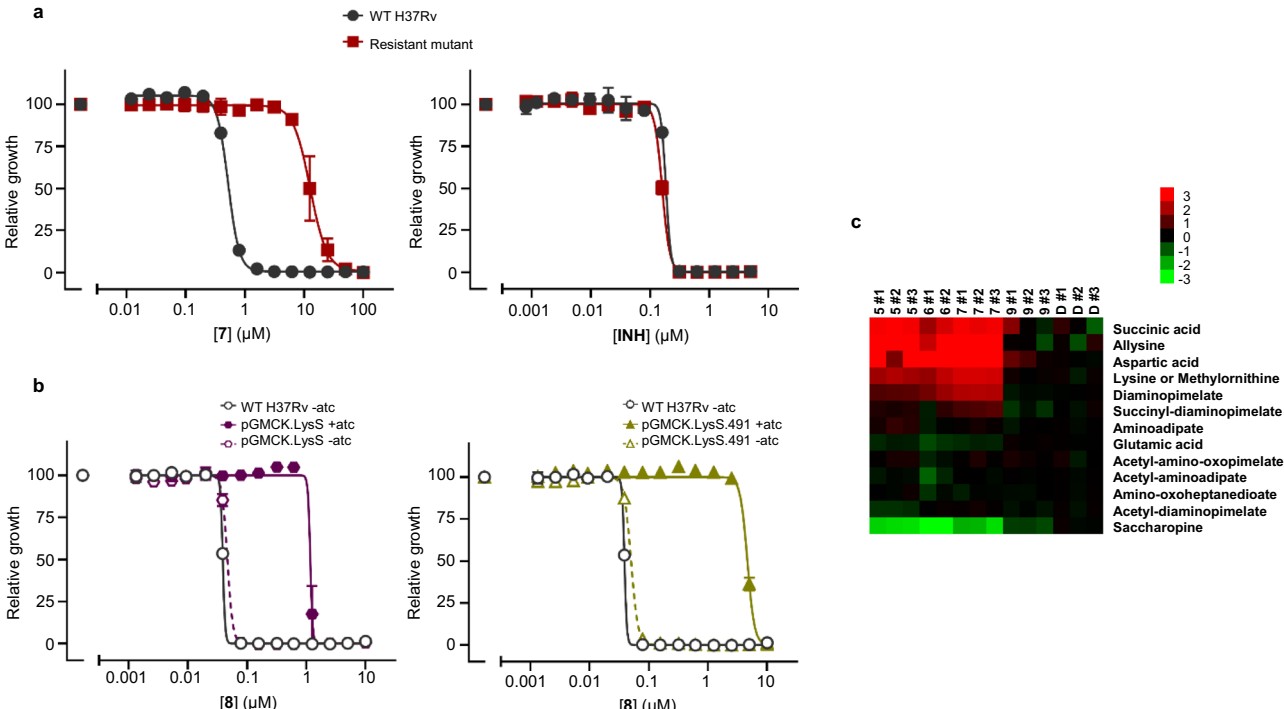

**Fig. 4 | Studies to explore the cellular mechanism of action. a** A resistant mutant was isolated against **7**, resistance was confirmed by looking at sensitivity over a concentration range compared to parental H37Rv for both **7** and the unrelated drug isoniazid. **b** Impact of TetON overexpression of either wild-type (WT) LysRS or mutant LysRS.491 on sensitivity to **8** ± anhydrotetracycline (ATc), for comparison the impact of **8** on H37Rv is also shown. For **a**, **b**, graphs are representative data from one of two independent experiments each run in triplicate presented as mean values ± standard deviation. **c** Metabolomic analysis of cells treated with three active and one inactive members of the LysRS inhibitor series (5× MIC) for 18 h. Metabolites related to the lysine biosynthetic pathway are shown. Drug concentrations were evaluated in triplicate per experiment, and the results are representative of at least two independent experiments. Data are depicted on a log$_2$ transformed ratio of ion intensity of metabolite abundance in treated vs untreated *M. tuberculosis* and displayed using the image generation program treeview (http://jtreeview.sourceforge.net/). Source data are provided as a Source Data file.

inhibit the KARS1 enzyme activity suggested that these compounds should not impact MPS. To explore this, an MPS assay was performed in HepG2 cells (Supplementary Table 5). The assay compares the production of two mitochondrial proteins, COX1, synthesized by mitochondrial ribosomes, and SDHA, synthesized by cytoplasmic ribosomes. Linezolid (and another antibacterial translation inhibitor, chloramphenicol) exhibited selective inhibition of COX1 production indicating inhibition of MPS. The early, non-selective molecule **2**, blocked expression of both COX1 and SDHA, indicating inhibition of both cytoplasmic and mitochondrial translation, whereas treatment with **7**, **5**, or **8** (≤100 μM) did not inhibit the production of either COX1 or SDHA. Thus, the more advanced LysRS inhibitors appear to avoid the cause of linezolid's major clinical liability.

When LysRS inhibitors were profiled against clinically relevant bacteria, they were relatively selective for *M. tuberculosis*, having no activity against Gram-negative bacteria and *Staphylococcus aureus* and only modest activity against other Gram-positive strains (Supplementary Table 6). This is in contrast to oxazolidinones which are active against most Gram-positive bacteria that cause disease[14]. As a translation inhibitor, linezolid has a largely bacteriostatic effect on growth in vitro[14,20] and time-kill assays demonstrated that **5** also had a bacteriostatic effect on *M. tuberculosis* growth in vitro (Supplementary Figure 3).

One important consideration for new tuberculosis drugs is whether there is pre-existing resistance in circulating clinical isolates. To address this concern, **5** and **8** were tested against 31 clinical isolates. Of these isolates, 30 had resistance to some standard clinical agents with 20 of them being either MDR+ or XDR. Both compounds were potent against all the resistant clinical strains, suggesting no pre-existing resistance towards this class of inhibitors among circulating clinical

strains (Supplementary Table 8). In addition, some laboratory-derived strains, resistant to either bedaquiline or pretomanid were evaluated for cross-resistance but none was seen (Supplementary Table 7).

In summary, DDD02049209 (**8**) is a candidate for development as tuberculosis therapeutic with potent in vivo activity and excellent drug-like properties. DDD02049209 is a non-covalent inhibitor of *M. tuberculosis* lysyl-tRNA synthetase and has progressed into early toxicity studies prior to the declaration as a preclinical development candidate.

## Methods

All animal protocols were approved by the Ethical Review Committee at the appropriate institution including the UoD, WuXi, GSK, and JHU.

### Chemical synthesis

see Supplementary Information.

### Biochemistry

**Protein expression and purification method for recombinant *M. tuberculosis* LysRS.** Synthetic DNA, codon optimized for expression in *E. coli*, encoding the full-length *M. tuberculosis* LysRS protein was synthesized (Genscript) and subcloned into a modified pET15b vector, with an N-terminal His tag and a Tobacco Etch Virus cleavage site. The resulting plasmid was transformed into *E. coli* BL21(DE3) (Stratagene) for expression. A starter culture was grown at 37 °C overnight, 10 mL was used to inoculate 1 L of LB Autoinduction media which was then grown for 48 h at 20 °C. Cell pellets were re-suspended in buffer containing 100 mM HEPES, 150 mM NaCl, 20 mM imidazole, 5% glycerol, pH 7.5, and lysed using a continuous flow cell disrupter (Constant Systems). The protein was purified using Ni affinity chromatography

followed by size exclusion chromatography, with the tag intact. Protein samples were concentrated to ~20 mg/mL in buffer containing 100 mM HEPES, 150 mM NaCl, 5% glycerol, pH 7.5, and stored at −80 °C.

**Protein expression and purification method for recombinant human KARS1.** Synthetic DNA, codon optimized for expression in *E. coli*, encoding full-length KARS1 was synthesized by Genscript and subcloned into a modified pET15b vector, with an N-terminal His tag and an HRV 3 C protease site. The resulting plasmid was transformed into *E. coli* BL21(DE3) (Stratagene) for expression. A starter culture was grown at 37 °C overnight, 10 mL was used to inoculate 1 L of LB Autoinduction media which was grown for 3 h at 37 °C and a further 20 h at 20 °C. Purification was similar to LysRS, using buffer consisting of 25 mM HEPES, 500 mM NaCl, 10% glycerol, 2 mM DTT, 20 mM imidazole pH 7.0 for the initial Ni affinity step and 25 mM HEPES, 500 mM NaCl, 5% glycerol, 2 mM DTT, pH 7.0 for size exclusion chromatography. Protein samples were stored at −80 °C without concentration, at approximately 1 mg/mL.

**M. tuberculosis LysRS in vitro enzyme assay and counter screen.** Potency of compounds was determined in 10 points dose-response inhibition curves. Compounds were put into 384-well clear-bottom plates (Greiner Bio-One, 781101) using an Echo 550 acoustic dispenser (Labcyte). LysRS reactions were started by adding enzyme into wells containing substrate and compounds using a Thermo Scientific Matrix Wellmate. LysRS assays were run in 50 μL reactions (Final concentration: 30 mM Tris-HCl 8.0, 40 mM $MgCl_2$, 140 mM NaCl, 30 mM KCl, 0.01% Brij-35, 1 mM DTT, 30 μM ATP, 12 μM lysine, 40 nM LysRS and 0.5 U/mL Pyrophosphatase) at room temperature for 20 h. Reactions were stopped by the addition of Biomol green (50 μL: Enzo Life Sciences) with the amount of free phosphate detected by measuring absorbance (650 nm) after 20 min further incubation (BMG Pherastar plate reader). Complete inhibition control reactions were performed in the absence of lysine as substrate. The screen used initially, when compound **1** was first identified, was based on the above assay with the following modification 200 nM LysRS, 3 μM ATP, 12 μM Lysine with an 8 h incubation. As compound potency increased the assay was modified to a more stringent assay. Counter screen assays to eliminate direct inhibition of pyrophsphatase were performed using 0.5 U/mL pyrophosphatase and 2 μM pyrophosphate as substrate in 100 mM Tris-HCl pH 7.6 buffer containing 1 mM MgCl2 and 1 mM TCEP for 2 h at room temperature. Biomol green was added and incubated for 20 min before detection as above. Samples were all run in duplicate and data was processed and analyzed using ActivityBase (IDBS).

**Human KARS1 in vitro assay.** This assay was done as for the LysRS assay except the final assay conditions were 30 mM Tris-HCl 8.0, 40 mM $MgCl_2$, 140 mM NaCl, 30 mM KCl, 0.01% Brij-35, 1 mM DTT, 3.5 μM ATP, 6 μM lysine, 200 nM KARS1 and 0.5 U/mL Pyrophosphatase, incubations were at room temperature for 5 h.

**Mutated LysRS protein expression and assay.** A synthetic mutant LysRS protein expression construct was prepared as described for the WT enzyme, except for the insertion of an AGC triplet to encode an extra Ser at residue 491. Thus, the final 26 residues were RLLMSLTGLS**S**IRETVLFPIVRPHSN rather than RLLMSLTGLSIR-ETVLFPIVRPHSN. The mutant protein was expressed in the same way as the WT enzyme and the in vitro enzyme assay was also performed as WT.

**Crystallography**
**M. tuberculosis LysRS crystallization.** Sitting drop vapor diffusion experiments using sparse matrix screens (Qiagen) gave an initial hit condition consisting of 0.2 M sodium acetate and 20% w/v PEG 3350.

For soaking experiments, crystals were grown using vapor diffusion in hanging drops, with a reservoir containing 0.25 M NaOAc and 14% w/v PEG 3350, with crystallization drops consisting of 1 μL of protein solution and 1 μL of reservoir solution. Prior to crystallization the protein, in the storage buffer, was incubated with a final concentration of 5 mM lysine. For soaking, crystals were transferred into drops consisting of 1 μL of the reservoir and 1 μL of storage buffer supplemented with 10 mM compound. Crystals were harvested after 1 hour of soaking, cryoprotected using reservoir solution supplemented with 33% glycerol, and flash frozen.

**M. tuberculosis LysRS crystal structure and refinement.** The initial structure of *M. tuberculosis* LysRS was solved through molecular replacement using the structure of *Cryptosporidium parvum* LysRS (PDB: 5elo) as the search model in Phaser[21]. Manual model building was performed using Coot[22] and the structure was refined using Refmac[23], incorporated into the CCP4 suite of software[24]. Ligand dictionaries were prepared using AceDRG[25], and model quality was assessed using Molprobity[26]. Lysine-only data were collected at Diamond Light Source using beamline I04 at a wavelength of 0.9795 Å, **2** co-crystal data at Diamond Light Source at beamline I04-1 at a wavelength of 0.91587 Å, and the data for the co-crystal with **8** were collected at beamline Proxima2A at SOLEIL at a wavelength of 0.98011 Å. Refinement statistics are given in Supplementary Table 9.

**Cell biology**
**M. tuberculosis phenotypic growth screen and assays.** The Global Health Chemical Diversity Library has been described briefly before[27]. This ~70,000 compound library was screened in a 3 and 4-day assay on glucose and dipalmitoylphosphatidyl choline-based media respectively essentially as described[28]. Minimum inhibitory concentrations were determined using either a previously published resazurin-based method[29] or an alternative optical density (OD) based method in which *M. tuberculosis* H37Rv was cultured in Middlebrook 7H9 medium supplied with 10% ADC and 0.025% Tyloxapol, then incubated at 37 °C for ~10 days. Following a purity check, subculture was performed in Middlebrook 7H9 medium supplied with 10% ADC and 0.025% Tyloxapol up to OD (600 nm) = 0.01 and incubated at 37 °C 4–6 days. By measuring the OD, the inoculum was adjusted to OD (600 nm) = 0.00125 which is equivalent to $1 \times 10^5$ cfu/mL. 50 μL of the inoculum was dispensed to wells already containing compound dissolved in DMSO (384 plates). Plates were placed in a sealed box to prevent evaporation and incubated at 37 °C for 8 days. The lids were removed, and the OD was determined using an EnVision plate reader (590 nm).

**Intramacrophage M. tuberculosis growth assays.** This assay was done essentially as previously described[29]. Human THP1 Monocytes have been shown as a good model to study the intracellular stages of *M. tuberculosis*. The assay uses a luminescent strain of *M. tuberculosis* to infect the monocytes, and luciferase activity is then measured using a commercial reagent (Bright-Glo™). The reagent causes cell lysis and generates a luminescent signal which is proportional to the activity of luciferase present which in turn is proportional to the number of viable bacteria in the culture. A culture of *M. tuberculosis* H37Rv:*luc* was grown in 7H9-ADC-Tyloxapol until the OD (600 nm) was 0.5–0.8. Cells were collected, dispersed with glass beads and re-suspended in RPMI1640 medium. THP1 Monocytes (ATCC TIB-202) were maintained in complete RPMI and incubated at 37 °C with 5% $CO_2$. THP1 Monocytes ($4 \times 10^5$ cell/mL) were infected (multiplicity of infection of 1) for 4 h in a roller bottle in RPMI + PMA. Excess bacteria were removed by washing and then 10,000 cells/well of infected THP1 cells were dispensed into white 384-well plates containing pre-plated compound dilutions. Cultures were incubated for 5 days at 37 °C 5% $CO_2$. At the end of the experiment, reconstituted Bright-Glo™ reagent (25 μL) was

added to each well, incubated at RT for a further 30 min before luminescence was detected.

**M. tuberculosis cidality assays.** Compounds were added at 10× MIC to a 10 mL exponential-phase culture of M. tuberculosis H37Rv (~5 × 10^5 CFU/mL) in Middlebrook 7H9 medium with 10% (vol/vol) OADC supplement and 0.05% (vol/vol) Tween 80. At the specified time points, aliquots of cultures were withdrawn, serially diluted, and plated onto a solid culture medium. Plates were then incubated at 37 °C, and CFU was counted after 3–4 weeks.

**Assessment against drug-resistant clinical isolates of M. tuberculosis.** The BACTEC MGIT 960 System (Becton Dickinson) was used for MIC determination in clinical isolates following the manufacturer's instructions.

**Assessment of general antibacterial activity.** Whole-cell antimicrobial activity was determined by broth microdilution using the Clinical and Laboratory Standards Institute (CLSI) recommended procedure, Document M7-A7. Compounds have been evaluated against a panel of Gram-positive and Gram-negative organisms, including *Enterococcus faecium*, *Enterococcus faecalis*, *Haemophilus influenzae*, *Moraxella catarrhalis*, *Streptococcus pneumoniae*, *E. coli*, and *Streptococcus pyogenes*. The MIC was determined as the lowest concentration of compound producing a reduction in the observed fluorescence higher than 80%.

**Cytotoxicity assays using human HepG2 cells.** Compound dilution curves were plated directly using a Labcyte Echo 550 acoustic dispenser (125 nL) in 384-well white clear-bottomed plates (Greiner). HepG2 cells (ECACC 85011430) were cultured in minimum essential medium (supplemented with glutamax) with 10% FCS and plated (25 μL) using a WellMate dispenser (1 × 10^5 per well) and incubated for 72 h. Doxorubicin was used as a positive control drug. Resazurin was then added to each well at a final concentration of 45 μM, and fluorescence was measured using PHERAstar LS (BMG Labtech) after 4 h of further incubation (excitation of 528 nm and emission of 590 nm). Raw data were normalized to controls and expressed as % growth. IC$_{50}$ was defined as the compound concentration that resulted in 50% inhibition.

## Mode of action studies

### Generation of M. tuberculosis resistant mutant to Compound 7

**Resistant Mutant isolation.** Approximately 4 × 10^9 CFU of WT H37Rv were plated onto 7H10 agar (10% [v/v] OADC, 0.5% [v/v] glycerol) plates containing 10 μM Compound 7 (~10 × MIC). After incubation at 37 °C for 9 weeks, one colony was isolated and propagated for further study. The initial number of bacteria was determined by plating dilutions of the original culture.

**Chromosomal DNA isolation.** Chromosomal DNA was isolated a protocol adapted from von Soolingen *et al*.[30]. Briefly, mid-exponential growth bacteria were suspended to a final OD$_{580}$ of 0.1–0.2 in 20 mL fresh 7H9 [4.7 g/L Middlebrook 7H9 Broth Base, 2.5 g/L Bovine Serum Albumin Fraction V protease-free (Roche), 1 g/L dextrose, 0.425 g/L NaCl, 0.2% (v/v) glycerol, and 0.05% (v/v) Tyloxapol] in a vented T75 flask, and were incubated for 7 d at 37 °C and 5% CO$_2$ in a humidified incubator. Bacteria were pelleted by centrifugation. The pellet was re-suspended in 1 mL TE Buffer [10 mM Tris, 1 mM EDTA, pH 8.0]. 50 μL of 10 mg/mL lysozyme (Sigma) prepared in TE Buffer and 0.25 μL of 100 mg/mL RNaseA (Qiagen) was added to the suspension, which was then briefly vortexed. The suspension was distributed equally between two 1.5-mL Eppendorf tubes and was incubated at 37 °C overnight. 5 μL of 20 mg/mL Proteinase K (Invitrogen) and 45 μL of 10% (w/v) SDS

solution was added to each tube followed by a brief vortex. The tubes were incubated at 65 °C for 1 h. 50 μL of 3 M NaCl and 40 μL of 10% (w/v) CTAB (hexadecyltrimethyl ammonium bromide) were added to each tube, followed by a brief vortex. The tubes were further incubated at 65 °C for 1 h. The contents of the two tubes derived from the same suspended pellet were re-combined in a single 2-mL Eppendorf tube. Ice-cold chloroform (700 μL) was added to the tube which was then shaken to mix the contents. The layers were separated by centrifugation in a microcentrifuge at 15,871 × *g* and 4 °C for 5 min. The upper aqueous layer was transferred to a new 2 mL tube. 1 mL of phenol–chloroform solution (from phenol:chloroform:isoamyl alcohol 25:24:1, saturated with 10 mM Tris, pH 8.0, 1 mM EDTA [Sigma]) was added to the tube which was then shaken to mix the contents well. The layers were separated by centrifugation in a microcentrifuge at 15,871 × *g* and 4 °C for 5 min. The upper aqueous layer was transferred to a new 2 mL tube and was mixed with 1 mL of chloroform. The layers were separated by centrifugation in a microcentrifuge at 15,871 × *g* and 4 °C for 5 min. The upper aqueous layer was transferred to a new 1.5 mL tube. Isopropanol (420 μL) was added to the tube and the solution was mixed by inversion to precipitate out the DNA. Sodium acetate (10 μL of 3 M) was added to the tube with mixing by inversion, and the DNA was pelleted by centrifugation in a microcentrifuge at 15,871 × *g* and 4 °C for 30 min. The supernatant was removed from the pellet, and the pellet was then washed with 500 μL of 75% (v/v) ethanol. The DNA was pelleted by centrifugation in a microcentrifuge at 15,871 × *g* and 4 °C for 5 min. The supernatant was removed, and the pellet was allowed to air dry. The DNA was re-suspended in 50 μL of TE Buffer.

**Whole-genome sequencing/analysis.** Approximately 200 ng of genomic DNA was sheared acoustically and HiSeq sequencing libraries were prepared using the KAPA Hyper Prep Kit (Roche). PCR amplification of the libraries was carried out for 10 cycles. 5–10 × 10^6 50 bp paired-end reads were obtained for each sample on an Illumina HiSeq 2500 using the TruSeq SBS Kit v3 (Illumina). Post-run demultiplexing and adapter removal were performed and fastq files were inspected using fastqc[31]. Trimmed fastq files were then aligned to the reference genome (*M. tuberculosis* str. H37RvCO; NCBI Reference Sequence: NZ_CM001515.1) using bwa mem[32]. Bam files were sorted and merged using samtools[33]. Read groups were added and bam files de-duplicated using Picard tools and GATK best-practices were followed for SNP and indel detection[34].

**Sensitivity of M. tuberculosis mutant resistant to LysRS inhibitor 7.** The strain was cultured in 10 mL of Middlebrook 7H9 supplemented with 0.2% (v/v) glycerol, 0.05% (v/v) Tyloxapol, and ADNaCl (0.5% [w/v] BSA, 0.2% [w/v] dextrose, and 0.85% [w/v] NaCl) in a 25 cm^2 tissue culture flask with a vented cap. After approx. 7 d at 37 °C and 5% CO$_2$ in a humidified incubator, growing to mid-log to log phase, each of the cultures were washed with fresh media and suspended to a final OD$_{580nm}$ of 0.01. Compounds were solubilized in DMSO and dispensed into black, clear-bottom 384-well tissue culture plates using an HP D300e Digital Dispenser as 14-point, twofold dilution series in triplicate. 50 μL of OD$_{580}$ 0.01 suspension was pipetted to each well and cultures were incubated for 7–14 d at 37 °C in the same conditions as above. Final OD$_{580nm}$ values were normalized to no-drug control wells.

**Generation of M. tuberculosis strains overexpressing WT and mutant LysRS.** Expression plasmids for WT LysRS (pGMCK.LysRS or the LysRS S491 mutant (pGMCK.LysRS.491) were cloned by gateway recombination as described previously[35]. Both plasmids contain *lysS* gene under the control of a tetracycline-inducible promoter, bear a kanamycin resistance cassette, and stably transform into Mtb via integration into the attachment site of the phage L5 (attL5).

**Growth inhibition comparisons for *M. tuberculosis* overexpressing strains.** As with the resistant mutant strain, except kanamycin at 25 μg/mL was included in growth media to maintain the overexpressing plasmid, and experiments were carried out ±anhydrotetracycline [ATc] at 500 ng/mL to induce overexpression.

**Metabolomic studies on *M. tuberculosis* treated with LysRS inhibitors.** Metabolomic analysis was carried out essentially as described previously[36,37]. In short, *M. tuberculosis* was grown in liquid 7H9 to an OD (580 nm) of 1, bacterial culture (1 mL) was then inoculated onto 0.22 mm nitrocellulose filter using vacuum filtration and placed on to 7H10 agar supplemented with 0.2% glucose. Inoculated filters were incubated at 37 °C for 5 days to generate sufficient biomass for subsequent metabolomic profiling. Bacteria grown on filters were transferred to Middlebrook 7H10 plates containing DMSO control or compounds at 5× MIC and incubated at 37 °C for 18 h. Bacterial metabolism was quenched by immersing filters into a solution of acetonitrile:methanol:water (40:40:20) precooled on dry-ice. Bacteria removed from these filters were mechanically lysed under continuous cooling at 2 °C, clarified by centrifugation, and the resultant supernatant filtered through a 0.22-mm filter. The biomass of each sample was determined by measuring residual protein content using a colorimetric assay (Pierce BCA Protein Assay). Metabolites were separated using an Agilent 1200 LC system with a Cogent Diamond Hydride Type C column using a gradient of water and acetonitrile with 0.2% formic acid. This system was coupled to an Agilent Accurate Mass 6220 TOF mass spectrometer, and detected ions were classified as metabolites based on unique accurate mass–retention time identifiers for masses showing the expected distribution of accompanying isotopomers. Metabolites were identified using Agilent Quantitative Analysis software with a mass tolerance of <0.005 Da. Metabolite abundances were reported in relation to untreated vehicle control samples.

**Mitochondrial protein synthesis inhibition assays.** HepG2 cells (ATCC HB-8065) were grown at 37 ˚C, 5% CO_2, 90% relative humidity to 60% confluence in T175 flasks in a growth medium consisting of DMEM containing 4.5 g/L glucose and 4 mM ʟ-glutamine (Gibco #11965-092) supplemented with 10% FBS and 1 mM sodium pyruvate. Cells were trypsinized in 0.05% trypsin-EDTA (Gibco) and re-suspended in 10 mL growth medium. Cells were filtered through a 40 μm filter and seeded in collagen-coated 96-well plates (Life technologies #A11428-03) at 3500 cells/well in a volume of 170 μL/well with plates agitated upon cell addition to ensure even distribution. After overnight incubation, 30 μL of growth medium with serially diluted compound ensuring a duplicate 10-point twofold titration series was added to each well ensuring that the final DMSO concentration did not exceed 0.25% with duplicate drug-free vehicle control wells as well as drug-free controls in duplicate for the background as detailed below. Chloramphenicol and linezolid were used as positive controls. The plates were incubated an additional 72 h after which the medium was gently aspirated and replaced with 4% paraformaldehyde (100 μL/well) prepared in Dulbecco's PBS. After 20 min fixation, the medium was removed, and wells washed three times with 200 μL/well PBS. Wells were treated for 5 min with 0.5% acetic acid (100 μL/well), wells were washed with 200 μL/well PBS and subsequently treated with permeabilization buffer (0.1 % Triton X-100) (100 μL/well) for 30 min. Buffer was removed and replaced with 2× blocking buffer from the ab110217 MitoBiogenesis™ In-Cell ELISA Kit (Abcam). After 2 h incubation, primary antibody was added (100 μl/well) in 1× blocking buffer per manufacturer's recommendation except for the background control wells which received 100 μl/well 1× blocking buffer only. Plates were incubated overnight at 4 °C after which medium was removed and wells washed three times at 200 μl/well with wash buffer (0.25% Tween-20 in PBS). Alkaline phosphatase/horse radish peroxidase-labeled secondary antibody solution

prepared according to the manufacturer's instructions in 1× blocking buffer was added at 100 μl/well to all wells. After 1 h incubation, wells were washed 4 times with wash buffer (200 μl/well) followed by addition of 100 μl/well freshly prepared per manufacturer's recommendation alkaline peroxidase development solution. The reaction was monitored at 405 nm in a Clariostar microplate reader (BMG Labtech). The reagent was removed, and 100 μl/well horse radish peroxidase development solution added. The reaction was monitored at 595 nm in the microplate reader after 15 min incubation. The development solution was completely removed, and plates blotted to dryness after which 1× Janus green stain (50 μl/well) was added. Plates were washed 5 times with water after 5 min, 0.5 M HCl (100 μl/well) added, incubated for 10 min with shaking and absorbance recorded at 595 nm. Data was interpreted per manufacturer's instructions and plotted in GraphPad Prism to calculate IC_{20} and IC_{50} values for Cox-I/SDHA and cell viability.

## Pharmacokinetic and efficacy studies

**Mouse pharmacokinetics.** All regulated procedures, at the University of Dundee, on living animals was carried out under the authority of a project licence issued by the Home Office under the Animals (Scientific Procedures) Act 1986, as amended in 2012 (and in compliance with EU Directive EU/2010/63). Licence applications will have been approved by the University's Ethical Review Committee (ERC) before submission to the Home Office. The ERC has a general remit to develop and oversee policy on all aspects of the use of animals on university premises and is a subcommittee of the University Court, its highest governing body. Test compound was dosed as a bolus solution intravenously (IV) at 3 mg/kg (dose volume:5 mL/kg; dose vehicle: 10% DMSO; 40% PEG 400; 50% Milli-Q water) or dosed orally (PO) by gavage as a suspension at 10 mg/kg (dose volume: 10 mL/kg; dose vehicle: 1% CMC to female C57/BLJ mice ($n$ = 3/dose route). Blood samples were taken from each mouse at 5, 15, 30 min, 1, 2, 4, 6, 8 h mixed with nine volumes of Milli-Q water and stored frozen until UPLC/MS/MS analysis. Pharmacokinetic parameters were derived from the blood concentration-time curve using PK solutions software v2.0 (Summit Research Services, USA).

**Rat pharmacokinetics.** As for mice except using male Hans Wistar rats (IV dose volume 1 mL/kg; PO dose volume: 5 mL/kg).

**Dog pharmacokinetics.** All regulated procedures were in accordance with the WuXi Institutional Animal Care and Use Committee (IACUC) standard animal procedures along with the IACUC guidelines that are in compliance with the Animal Welfare Act, and the Guide for the Care and Use of Laboratory Animals. As for mouse except using male beagle dogs (IV dose volume: 1 mL/kg; dose vehicle: 5% DMSO; 40% PEG 400; 55% Milli-Q water; PO dose volume: 5 mL/kg). Pharmacokinetic parameters were derived from the blood concentration-time curve using WinNonLin Professional Software v 6.3 (Pharsight Corporation).

**In vivo murine models of acute *M. tuberculosis* infection.** All procedures were performed in accordance with protocols approved by the GSK Institutional Animal Care and Use Committee and met or exceeded the standards of the American Association for the Accreditation of Laboratory Animal Care (AAALAC). All animal studies were ethically reviewed and carried out in accordance with European Directive 2010/63/EEC and the GSK Policy on the Care, Welfare, and Treatment of Animals. Specific pathogen-free, 8–10 week-old female C57BL/6 mice were purchased from Harlan Laboratories and were allowed to acclimate for one week. Mice were housed in ambient temperature (22 ± 2 °C), 55 ± 15% relative humidity, and 12 h dark/light cycle. Mice were allowed to acclimatize for one week. Mice are placed in cages in the ABSL-3 facility and provided food and water ad libitum. The experimental design for the acute assay has been previously

described[38]. In brief, mice were intratracheally infected with 100,000 CFU/mouse (*M. tuberculosis* H37Rv strain). Products were administered for 8 consecutive days starting one day after infection (1mouse/dose). To determine the CFU counts, lung homogenates were plated in 10% OADC-7H11 medium supplemented with activated charcoal (0.4%) for 18 days at 37 °C. For a dose/response 10 mice are used per compound, each one is administered a different dose and all the data are fitted to a sigmoidal curve (GraphPad, Prism®).

**In vivo murine models of chronic *M. tuberculosis* infection.** As the above studies were approved at GSK. Specific pathogen-free, 8–10 week-old female C57BL/6 mice were purchased from Harlan Laboratories and were allowed to acclimate for one week in housing conditions as described above. The experimental design for the chronic assay has been previously described[39]. Mice were intratracheally infected with 100 CFU/mouse and the products were administered daily (7 days a week) for 8 consecutive weeks starting 6 weeks after infection (2 animals/dose). Lungs were harvested 24 h after the last administration. All lung lobes were aseptically removed, homogenized, and frozen. To determine the CFU counts, homogenates were plated in 10% OADC-7H11 medium supplemented with activated charcoal (0.4%) for 18 days at 37 °C. For the dose/response, 20 mice are used per compound (two mice per dose) and all the data are fitted to a sigmoidal curve (GraphPad, Prism®).

**Combination studies in murine models of subacute *M. tuberculosis* infection.** All procedures involving mice at Johns Hopkins University were approved by the Johns Hopkins University Animal Care and Use Committee and were conducted with strict adherence to the Animal Welfare Act and Public Health Service Policy. The Johns Hopkins University is accredited by the private Association for the Assessment and Accreditation of Laboratory Animal Care International. Mice were housed in ambient temperature (20–23 °C), 30–70% humidity, and 12 h dark/light cycle in microtainer cages in an ABSL-3 facility and provided food and water ad libitum. Female BALB/c mice (4–6 weeks old) were aerosol-infected with 4.19 $\log_{10}$ CFU of *M. tuberculosis* H37Rv on D-14. Treatment started 2 weeks later (D0) and lasted for 2 months (8 weeks). Untreated mice were sacrificed for lung CFU counts on D-13 and on D0 to determine the number of CFU implanted and the number present at the start of treatment, respectively. Doses (mg/kg) tested were bedaquiline (B: 25 mg/kg, once daily), pretomanid (Pa: 25 mg/kg, bid, i.e., ~8 hrs apart), **5** (K: 50 and 100 mg/kg twice daily). K was formulated in 1% methylcellulose in distilled water. B was prepared in an acidified 20% hydroxypropyl-β-cyclodextrin (HPCD, Sigma) solution every 2 weeks. Pa was prepared in CM-2 (cyclodextrin micelle, containing 10% HPCD and 10% lecithin, ICN, Aurora, OH) monthly and diluted each week. The volume of administration was 0.2 mL. Mice weighed ~20 g. Drugs were administered 5 days per week. Efficacy determinations were based on lung CFU counts after 2 months of treatment. To determine CFU counts, lung homogenates were plated in serial 10-fold dilutions on selective Middlebrook 7H11 plates supplemented with 10% OADC containing charcoal (0.4%) to prevent drug carryover. Group means were compared using one-way ANOVA with Dunnett's or Bonferroni's post-test, as appropriate, to control for multiple comparisons using GraphPad Prism. Error bars are shown for the standard deviation within each group.

**DMPK/safety assays**
**Aqueous solubility.** Aqueous kinetic solubility was assessed using an "in-house" developed method. Test compounds were dissolved in DMSO to give 10 mM stock solutions. Solubility test samples were prepared by adding (5 μL) stock to PBS pH 7.4 (195 μL). This solution was then mixed for 24 h (rotary mixing, 900 rpm, 25 °C) excluding light. After mixing, the solubility test samples were vacuum filtered (Millipore Multiscreen HTS filter, 96-well format) to remove any undissolved material. The filtrate was analyzed for dissolved drug compound using a truncated UHPLC methodology. A calibration solution was prepared for each test compound. The 10 mM stock solution was diluted in DMSO to give a 500 μM solution. This solution was further diluted with 50:50 acetonitrile:water to give a 50 μM solution. Aliquots (0.2, 2.0, and 5.0 μL) of this 50 μM solution were then injected onto the UHPLC system, and the areas of the resultant peaks were integrated to produce a calibration line. Aliquots of the test sample filtrate (0.4 and 5.0 μL) were then injected onto the UHPLC system and the resultant test compound peak areas quantified using the calibration line.

**PAMPA permeability assay.** The permeability assay was performed as previously described[40] using a 96-well precoated BD Gentest PAMPA plate (BD Biosciences, U.K.). Each well was divided into two chambers: donor and acceptor, separated by a lipid-oil-lipid trilayer constructed in a porous filter. The effective permeability, Pe, of the compound was measured at pH 7.4. Stock solutions (5 mM) of the compound were prepared in DMSO. The compound was then further diluted to 10 μM in phosphate-buffered saline at pH 7.4. The final DMSO concentration did not exceed 5% v/v. The compound dissolved in phosphate-buffered saline was then added to the donor side of the membrane, and phosphate-buffered saline without the compound was added to the acceptor side. The PAMPA plate was left at room temperature for 5 h; after which time, an aliquot (100 μL) was then removed from both acceptor and donor compartments and mixed with acetonitrile (80 μL) containing an internal standard: donepezil at 50 ng/mL. The samples were centrifuged (10 min, 5 °C, 3270×*g*) to sediment precipitated protein and sealed prior to UPLC-MS/MS analysis using a Quattro Premier XE (Waters Corp, USA). Recovery of the compound from donor and acceptor wells was calculated, and data were only accepted when recovery exceeded 70%.

**Plasma protein binding assay.** This assay was done as previously described[40]. In brief, a 96-well equilibrium dialysis apparatus was used to determine the free fraction in plasma for each compound (HT Dialysis LLC, Gales Ferry, CT). Membranes (12–14 kDaA cutoff) were conditioned in deionized water for 60 min, followed by conditioning in 80:20 deionized water/ethanol overnight, and then rinsed in water and isotonic buffer before use. Plasma from appropriate species was removed from the freezer and allowed to thaw on the day of the experiment. Thawed plasma was then centrifuged (Allegra X12-R, Beckman Coulter, USA) and spiked with test compound (final concentration 10 μg/mL), and 150 μL aliquots (*n* = 3 replicate determinations) were loaded into the 96-well equilibrium dialysis plate. Dialysis against isotonic buffer (150 μL) was carried out for 5 h in a temperature-controlled incubator at ca. 37 °C (Barworld scientific Ltd., UK) using an orbital microplate shaker at 100 rpm (Barworld scientific Ltd., UK). At the end of the incubation period, 50 μL aliquots of plasma or buffer were transferred into a 96-well deep plate and the composition in each well was balanced with control fluid (50 μL), such that the volume of buffer to plasma is the same. Sample extraction was performed by the addition of 200 μL of acetonitrile containing an appropriate internal standard. Samples were allowed to mix for 1 min and then centrifuged at 3000 rpm in 96-well blocks for 15 min (Allegra X12-R, Beckman Coulter, USA) after which 150 μL of supernatant was removed to 50 μL of water. All samples were analyzed by UPLC-MS/MS on a Quattro Premier XE Mass Spectrometer (Waters Corporation, USA). The unbound fraction was determined as the ratio of the peak area in the buffer to that in plasma.

**Intrinsic clearance (cli) assays.** Compounds (0.5 μM) were incubated with female CD1 mouse, male Hans Wister rat, male beagle dog, or pooled human liver microsomes (Xenotech LLC; 0.5 mg/mL 50 mM potassium phosphate buffer, pH7.4) as previously described[41]. The

reaction started with the addition of excess NADPH (8 mg/mL 50 mM potassium phosphate buffer, pH7.4). Immediately, at time zero, then at 3, 6, 9, 15, and 30 min an aliquot (50 μL) of the incubation mixture was removed and mixed with acetonitrile (100 μL) to stop the reaction. The internal standard was added to all samples, the samples were centrifuged to sediment precipitated protein and the plates were then sealed prior to UPLC/MS/MS analysis using a Quattro Premier XE (Waters Corporation, USA). XLfit (IDBS, UK) was used to calculate the exponential decay and consequently the rate of intrinsic clearance (Cli). Verapamil (0.5 μM) was used as a positive control to confirm acceptable assay performance.

**Hepatocyte stability assays.** As previously described[41] cryopreserved hepatocytes from Rat Han Wistar, Mouse CD1, and Human, were supplied by ThermoFisher Scientific, and dog cryopreserved hepatocytes were supplied by Xenotech LLC. Cells were thawed according to the manufacturer's instructions and cell viability was determined using trypan-blue exclusion. Compounds were incubated at a final concentration of 0.5 μM with hepatocytes (0.7 million cells/mL), in suspension. Aliquots of incubation were removed at 0, 10, 20, 30, 45, 60, 75, 90, 105, and 120 min to a termination solution containing an appropriate internal standard. The ratio of compound to internal standard was measured by UPLC/MS/MS analysis. A positive control, 7-ethoxycoumarin was also evaluated to ensure metabolic activity of the cells.

**In vitro safety assessment assays.** Since compounds will be administered as a combination **5** & **8** were evaluated for CYP450 inhibition (isoforms CYP1A2, CYP2C9, CYP2C19, CYP2D6, and CYP3A4) and exhibited no direct inhibition ($IC_{50} > 100\,\mu M$) across the isoforms tested. Neither compound raised concerns in a 50 receptor off-target eXP panel (all $IC_{50} > 25\,\mu M$)[42]. Both compounds were clean in genotoxicity assessments: AMES test[43] and mouse lymphoma assay[44]. Both compounds showed no in vitro hERG channel inhibition ($IC_{50} > 50\,\mu M$) and were clean in a cardiomyocyte multi-electrode assay[45].

### Reporting summary
Further information on research design is available in the Nature Research Reporting Summary linked to this article.

## Data availability
The three crystal structures have been deposited in the protein data bank under the codes 7QH8, 7QHN, and 7QI8. The authors declare that data supporting the findings of this study are available within the paper and its supplementary information files. Source data are provided in this paper.

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

## Acknowledgements

We thank Lucy Ellis, Nicole Mutter, Fred Simeons, Yoko Shishikura, Liam Ferguson, Lorna Campbell, Alex Cookson, Kirsty Cookson, Desiree Zeller, and Kieran Cartmill, for technical assistance. Gail Louw provided strains resistant to bedaquiline or pretominid. This work was funded in part by an award to P.G.W. from the Bill and Melinda Gates Foundation (OPP1066891 and OPP1191579) and Wellcome Trust, (100195/Z/12/Z); awards to D.S. from B&MGF (INV-010616 and INV-004761); awards to K.Y.R. from B&MGF (INV-004709 and OPP 1177930); E.L.N. was supported by TB Alliance with funds from Australia Aid, B&MGF (OPP1129600), the Germany Federal Ministry of Education and Research through KfW, Global Health Innovative Technology Fund, Irish Aid, Netherlands Ministry of Foreign Affairs and UK Aid. Work was also funded in part, by the Intramural Research Program of the NIH, NIAID. We acknowledge the use of the Integrated Genomics Operation Core at MSKCC, funded by the NCI Cancer Center Support Grant (CCSG, P30 CA08748), Cycle for Survival, and the Marie-Josée and Henry R. Kravis Center for Molecular Oncology. The authors would like to thank Diamond Light Source (proposal mx14980; beamlines IO4 and IO4-1) and SOLEIL (proposal 20181042; beamline Proxima2A) for beamtime, and the staff at the beamlines for assistance with crystal testing and data collection. The authors would like to thank Peter Warner of the B&MGF for advice, support, and encouragement.

## Author contributions

L.C. coordinated the project, led the medicinal chemistry program, interpreted data, contributed to drug design, and synthesis. S.G. coordinated the project, led the biology team, designed biology experiments, interpreted data, and wrote the manuscript. S.H.D., M.M., and K.I.B. contributed to drug design and synthesis. B.B., I.G., L.E., R.B., and P.W. advised synthetic program. S.D., F.T., J.P., K.D., and S.S. developed and ran in vitro lysyl-tNRA assays. E.P-H., and M.R. performed in vitro antibacterial assays. D.R., J.T., and A.D. solved and refined LysRS crystal structures. C.J. and F.Z. performed silico drug design and crystallography. C.E., H.K, J.B., Q.W., V.T., H.B., and D.S. designed, performed, and interpreted resistant mutants and another cellular mode of action experiments. K.Y.R., and N.N. supervised and performed metabolomic analysis. I.C. performed in vitro safety assays. O.E., J.R., L.S., M.C., and K.D.R. supervised, designed and performed in vitro/in vivo ADME/PK studies. L.G-L., and P.C.C. designed and performed acute and chronic efficacy studies. P.J.C, S.L, Y.C. N.F., A.U., and E.N. designed and performed in vivo combination studies. L.E. and R.B. contributed to project coordination. P.W. supervised the program and obtained project funding. L.C, C.E., P.J.C., E.N., D.S., P.C.C., K.Y.R., H.B., R.B., S.H.D, and P.W. contributed to the manuscript. All authors approved this paper for publication.

## Competing interests

These authors are employees and/or shareholders in GlaxoSmithKline: E.P.H., M.J.R., L.G.L., P.C.C., I.C., K.D.R., L.E., R.H.B., and P.G.W. The other authors declare no competing interests.

## Additional information

[1]Drug Discovery Unit, Wellcome Centre for Anti-Infectives Research, Division of Biological Chemistry and Drug Discovery, College of Life Sciences, University of Dundee, Dundee DD1 5EH, UK. [2]Dept. of Microbiology and Immunology, Weill Cornell Medical College, New York, NY, USA. [3]Global Health Medicines R&D, GlaxoSmithKline, Severo Ochoa 2, Tres Cantos 28760 Madrid, Spain. [4]Sloan Kettering Institute, Memorial Sloan Kettering Cancer Center, 1275 York Avenue, New York, NY, USA. [5]Tuberculosis Research Section, Laboratory of Clinical Immunology and Microbiology, NIAID, NIH, 9000 Rockville Pike, Bethesda, MD, USA. [6]Center for Tuberculosis Research, Department of Medicine, Johns Hopkins University, School of Medicine, Baltimore, MD, USA. [7]Global Alliance for TB Drug Development, New York, NY, USA. ✉e-mail: l.a.t.cleghorn@dundee.ac.uk

