## [Peer review file · Nature Communications]

REVIEWER COMMENTS

Reviewer #1 (Remarks to the Author):

This manuscript describes impressive and important work on TB drug development, especially in light of recent recommendations of WHO regarding the treatment on MDR/RR-TB. The authors demonstrate that compound 8 is a powerful molecule with a novel mechanism of action, and shows activity in animal models of TB infection. I recommend the publication of this work provided that the authors answer the following points:

Major:

- Figure 1, extended figures 4 and 5. Standard deviation should be visible for all measured values as well as the number of biological replicates.
- High level resistance selected with compound 7 (30x MIC) displays only a moderate effect on compound 8 activity (3x MIC), the lead. This observation is overlooked in the manuscript and should be discussed structurally as is for differences between compound 2 and compounds 5 and 7.
- Given the aforementioned discrepancies, compound 8 must be tested in figure 4B system for better clarity: does overexpression of LysS or LysS.491 result in decreased activity of compound 8 as well?
- Authors convincingly suggest that compound 8 could be evaluated as a replacement of the notoriously toxic linezolid as a translation inhibitor in BPaL. However, no data is made available for cross resistance of compound 8 with Bedaquiline or Pretomanid. Extended Figure 5B must include MIC of compound 8 with resistant strains to Bedaquiline and Pretomanid.
- Extended Figure 4C : data points as well as error bars should be visible. The number of biological replicates should be stated in the legend.
- Extended figure 5 : description of the clinical strains must be included : lineage, origin.

Minor:

- Figure 3B: Authors should include data of compound 5 (K) alone in this model to allow better understanding and deconvolution of the combination results or state why they did not include this treatment group in the experiment.
- Figure 4B: legend is missing for presumably H37Rv - atc and H37Rv + atc

Regards,

Reviewer #2 (Remarks to the Author):

All round a well written paper with a compelling story, which the readers will enjoy reading. For this manuscript to be published the authors need to address the following small issues.

1. Need to reformat Figure 1 so it is nicer to read.

2. For the readers the authors need to note in the introduction that the gene for human lysyl-tRNA synthetase codes for both mitochondrial and cytoplasmic forms by alternative splicing of KARS1 gene. This is not the case for a lot of other aminoacyl-tRNA synthetases.

- This is important in the later MPS assays as the biochemistry is effectively looking at both cytoplasmic and mitochondrial enzymes.

3. The authors might want to use KARS1 rather than HsKRS, KARS1 have 125,000 Google hits while HsKRS top hit is a brand of Nike sneakers.

4. The authors should add a discussion on why the IC50 is close to the MIC as for a lot of AARS inhibitors that bind to the aminoacylation site the IC50 needs to be much lower to obtain an MIC.

- This could be due to a long residence time and being non-competitive with ATP or lysine.

- However, any enzymology experiments is outside the scope for the manuscript but it is highly recommended that the authors do eventually publish on this.

5. One experiment that should be done is a competitive MIC with lysine as one problem with aminoacylation inhibitors is that they are competitive with the enzymes native substrates for example mupirocin. Although it is unlikely from the binding mode but as MIC + lysine so easy to do it should be done.

6. The authors in line 125 need to define what clean is for off-target receptor panels, and genotoxicity and cardiotoxicity assays, this is too subjective.

7. In Figure 3, it would help the reader if the authors made the chronic model 3B as it is best to discuss monotherapy before through in the kitchen sink of combination therapy. Also the subacute combination therapy was performed in a different mouse strain which needs to be noted.

8. It is very sad that the authors didn't test GSK656 and this LysS inhibitor as a combination AARS inhibitors might have some unusual and synergist activity.

9. In the subacute BALB/c model do the authors have LysS + Bedaquiline efficacy data as just looking at linezolid replacement is short sighted.

10. There is a far amount of text addressing the problematic combination therapy in the BALB/c subacute model, this can be shortened to save space.

11. In figure 4, if you include compound 9 the authors should also include compounds 5, 6 and 7 with their LysS IC50 numbers for the ease of reading.

12. The authors need to state in figure 4 that D stands for DMSO.
13. In extended figure 4C the authors need to put in the time points for the time-kill.
14. In extended figure 5A about 5 (μM) and 8 (μM) the authors should write MIC like in figure 5B.

Reviewer #3 (Remarks to the Author):

IN this interesting article Green et al describe the development of a new series of LysRS inhibitors with good pharmacological properties, including the determination of several crystal structures that support the SBDD.

The article is well written and convincingly shows that the identified compounds are promising leads towards the development of new anti-microbial drugs targetting LysRS.

From the point of view of the analysis, I think that the manuscript would benefit from an evolutionary analysis of MtLysRS, or at least a description of the known phylogenetics of this particular enzyme. This would help readers understand, for example, why the lack of mitochondrial affectation is an advantage.

In my opinion, however, this manuscript lacks novelty and is better suited to a more specialized journal.

Also, the authors need to improve their consistency when referring to lysyl-tRNA synthetase, or aminoacyl-tRNA synthetases

in general (please correct the last line of conclusions). Genes coding for aminoacyl-tRNA synthetases are conventionally named

with the cognate amino acid and the letter S, all in cursive. For example, the gene coding for a lysyl-tRNA synthetase would be *LysS* (in italics). The enzymes themselves can be referred to, for example, as KRS, LysRS, or KARS. Please use a consistent nomenclature throughout the text.

Reviewer #4 (Remarks to the Author):

Well-written manuscript deserving of publication. This is an area of high unmet need. I agree with the target selection: the most successful drugs in TB have been protein synthesis inhibitors and drug candidates selected from this series have a higher likelihood of becoming part of a sterilising regimen. SAR is well described.

Minor comments

Why did the team choose to look for non-covalent inhibitors? A brief rationale for not looking for irreversible or suicide inhibitors would be helpful to the non-specialist reader.

The data against intra-cellular cultures of MTB is very interesting. Is there any data on intracellular concentrations?

Target modification is not the only potential resistance mechanism.

Resistance can also be achieved by other mechanisms: e.g., efflux pumps, metabolic bypass (how much redundancy is there in the genome?). A more thorough assessment of this is needed (a paper/in silico assessment would do).

Major comment

When searching for pre-existing resistance, I would want to know how globally representative the 31 clinical isolates were: for example, 31 isolates all collected from one hospital in Madrid or all from one hospital Dundee - that would be inappropriate.

3rd August 2022

We thank the reviewers for reviewing our manuscript entitled "Lysyl-tRNA synthetase a new target for urgently needed M. tuberculosis drugs". We thank them for their positive feedback and helpful comments/critiques. Below are point-by-point responses to their comments, I hope we have addressed everything to their satisfaction.

Yours sincerely,

Laura Cleghorn

Reviewer #1 (Remarks to the Author):

This manuscript describes impressive and important work on TB drug development, especially in light of recent recommendations of WHO regarding the treatment on MDR/RR-TB. The authors demonstrate that compound 8 is a powerful molecule with a novel mechanism of action, and shows activity in animal models of TB infection. I recommend the publication of this work provided that the authors answer the following points:

Major:

- Figure 1, extended figures 4 and 5. Standard deviation should be visible for all measured values as well as the number of biological replicates.

As suggested by reviewer 2, Figure 1 has been modified from a tabular format to a figure. The number of replicates and 95% confidence interval were already included in the text for compound 8 the preclinical development candidate. The number of replicates and the standard deviation for all the compounds will be included in the source data file but adding it to the figure makes it appear very cluttered (see below) but we can include it, if this is the preferred format. Likewise for Supplementary Figure 4 all data points and statistics are included in the source data file and error bars have been added to the graph in 4C.

- High level resistance selected with compound 7 (30x MIC) displays only a moderate effect on compound 8 activity (3x MIC), the lead. This observation is overlooked in the manuscript and should be discussed structurally as is for differences between compound 2 and compounds 5 and 7.

The text has been modified to now describe the impact of the mutation on compound 8

- Given the aforementioned discrepancies, compound 8 must be tested in figure 4B system for better clarity: does overexpression of LysS or LysS.491 result in decreased activity of compound 8 as well?

As suggested, we have tested compound 8 in both the overexpressing strains and still see a decreased activity. To avoid duplication of similar data we have replaced the new data for compound 8 in place of compound 7 in Figure 4B. We have reorganised the text in the manuscript around this section as well.

- Authors convincingly suggest that compound 8 could be evaluated as a replacement of the notoriously toxic linezolid as a translation inhibitor in BPaL. However, no data is made available for

cross resistance of compound 8 with Bedaquiline or Pretomanid. Extended Figure 5B must include MIC of compound 8 with resistant strains to Bedaquiline and Pretomanid

We have included an extra table (Extended data Figure 5C) which includes both H37Rv and clinical strains resistant to either bedaquiline or pretomanid. None of these strains show cross resistance to LysRS inhibitors and they involve multiple mechanisms of resistance to each drug.

- Extended Figure 4C : data points as well as error bars should be visible. The number of biological replicates should be stated in the legend.

We have redrawn the graph to make the data points visible and to include the error bars as suggested. We have also modified the legend to include the number of replicates.

- Extended figure 5 : description of the clinical strains must be included : lineage, origin.

Unfortunately, the lineage data is not available for these strains. These strains are all clinical isolates from Vall d'Hebron Hospital. The intention of Sup 5C was to show that “the standard mechanisms of resistance” to current clinical agents are unlikely to impact sensitivity to LysRS inhibitors; it was not to represent detailed profiling of global *M. tuberculosis* clinical strains. The legend has been modified to highlight the one source of the clinical isolates. Previously we had tested compound 7 against ~15 clinical samples from a lab in South Africa, including some with different lineages and resistance profiles, and again no impact was seen on sensitivity to 7. To include both sets of data would have been duplicative. More in-depth clinical isolate profiling will continue as Compound 8 progresses through preclinical development.

Minor:

- Figure 3B: Authors should include data of compound 5 (K) alone in this model to allow better understanding and deconvolution of the combination results or state why they did not include this treatment group in the experiment.

The text has been modified to explain the rationale for dose selection

- Figure 4B: legend is missing for presumably H37Rv - atc and H37Rv + atc

The figure and legend have been corrected to include this.

Reviewer #2 (Remarks to the Author):

All round a well written paper with a compelling story, which the readers will enjoy reading. For this manuscript to be published the authors need to address the following small issues.

1. Need to reformat Figure 1 so it is nicer to read.

Figure 1 was presented as a table primarily for space limitations. We have included a pictorial version of this table as an alternative format

2. For the readers the authors need to note in the introduction that the gene for human lysyl-tRNA synthetase codes for both mitochondrial and cytoplasmic forms by alternative splicing of KARS1 gene. This is not the case for a lot of other aminoacyl-tRNA synthetases.

•This is important in the later MPS assays as the biochemistry is effectively looking at both cytoplasmic and mitochondrial enzymes.

The text has been modified as suggested to explain why oxazolidinones have an impact on mitochondrial protein synthesis and LysRS inhibitors do not due to the absence of a mitochondrial encoded gene.

3. The authors might want to use KARS1 rather than HsKRS, KARS1 have 125,000 Google hits while HsKRS top hit is a brand of Nike sneakers.

As suggested, we have changed HsKRS to KARS1

4. The authors should add a discussion on why the IC₅₀ is close to the MIC as for a lot of AARS inhibitors that bind to the aminoacylation site the IC₅₀ needs to be much lower to obtain an MIC.

•This could be due to a long residence time and being non-competitive with ATP or lysine.

•However, any enzymology experiments is outside the stop for the manuscript but it is highly recommended that the authors do eventually publish on this.

We agree that detailed enzymology is beyond this “introduction overview” article for the identification of the preclinical lead. We do intend to follow this article with more in-depth manuscripts relating to different aspects of biology and chemistry for the series. We have included a brief section in the SAR part of the manuscript mentioning the equivalence of IC₅₀ and MIC.

5. One experiment that should be done is a competitive MIC with lysine as one problem with aminoacylation inhibitors is that they are competitive with the enzymes native substrates for example mupirocin. Although it is unlikely from the binding mode but as MIC + lysine so easy to do it should be done

We have performed this experiment and there was no impact on MIC (see below). We have mentioned this in the SAR section including the contrasting effect with mupirocin but left it as “data not shown” as it doesn’t significantly impact the message/conclusions of the manuscript.

		plus 1mM Lysine	
	MIC (uM)	MIC (uM)	Ratio (+/-)
5	1.01	0.87	0.9
8	0.08	0.08	1.0
Moxifloxacin	1.23	2.75	2.2
Isoniazid	3.20	2.72	0.9

6. The authors in line 125 need to define what clean is for off-target receptor panels, and genotoxicity and cardiotoxicity assays, this is too subjective.

This information is provided in the Sup info, although that was not indicated in the original draft. The text has now been modified to show the location of the definitions

7. In Figure 3, it would help the reader if the authors made the chronic model 3B as it is best to discuss monotherapy before through in the kitchen sink of combination therapy. Also the subacute combination therapy was performed in a different mouse strain which needs to be noted.

Figure and legend have been modified as suggested

8. It is very sad that the authors didn't test GSK656 and this LysS inhibitor as a combination AARS inhibitors might have some unusual and synergist activity.

We agree this would be a very interesting combination to explore. Unfortunately, in the early development of this series it was not possible to evaluate all the possible combinations of interest. A decision was made to focus on BPaL as this is a clinically proven combination, while GSK656 is only in early clinical development. Bedaquiline-sparing regimens will be a major focus for future combination studies with our KRS inhibitors, and 656 would be a candidate for those assessments.

9. In the subacute BALB/c model do the authors have LysS + Bedaquiline efficacy data as just looking at linezolid replacement is short sighted.

Unfortunately, the suggested treatment arm was not included in this early-stage combination study. The aim of this initial model was to determine whether the LysRS inhibitor could replace linezolid in the clinically approved combination since they are both translational inhibitors, but linezolid is known to have clinical toxicities. As the lead compound progresses further through preclinical development more extensive combination studies are envisaged evaluating a range of double, triple and quadruple drug cocktails.

10. There is a far amount of text addressing the problematic combination therapy in the BALB/c subacute model, this can be shortened to save space

We have truncated the text slightly and removed comments about the clinical issues of linezolid that were repeated elsewhere

11. In figure 4, if you include compound 9 the authors should also include compounds 5, 6 and 7 with their LysS IC50 numbers for the ease of reading.

Compound 9 was included in Fig 4 because it was not in Fig 1 (as were 5-7) as such it was the first mention of the compound. The structure is shown in the supplemental information, so we have removed it from the Fig 4.

12. The authors need to state in figure 4 that D stands for DMSO.

We have modified the text as suggested

13. In extended figure 4C the authors need to put in the time points for the time-kill

We have redrawn the graph to make the time points visible & to include the error bars as suggested.

14. In extend ended figure 5A about 5 (μ M) and 8 (μ M) the authors should write MIC like in figure 5B.

We have modified the table as suggested

Reviewer #3 (Remarks to the Author):

IN this interesting article Green et al describe the development of a new series of LysRS inhibitors with good pharmacological properties, including the determination of several crystal structures that support the SBDD.

The article is well written and convincingly shows that the identified compounds are promising leads towards the development of new anti-microbial drugs targeting LysRS.

From the point of view of the analysis, I think that the manuscript would benefit from an evolutionary analysis of MtLysRS, or at least a description of the known phylogenetics of this particular enzyme. This would help readers understand, for example, why the lack of mitochondrial affectation is an advantage.

The text has been modified as suggested to explain why oxazolidinones have an impact on mitochondrial protein synthesis and LysRS inhibitors do not due to the absence of a mitochondrial encoded gene.

In my opinion, however, this manuscript lacks novelty and is better suited to a more specialized journal.

Also, the authors need to improve their consistency when referring to lysyl-tRNA synthetase, or aminoacyl-tRNA synthetases in general (please correct the last line of conclusions). Genes coding for aminoacyl-tRNA synthetases are conventionally named with the cognate amino acid and the letter S, all in cursive. For example, the gene coding for a lysyl-tRNA synthetase would be *LysS* (in italics). The enzymes themselves can be referred to, for example, as KRS, LysRS, or KARS. Please use a consistent nomenclature throughout the text

We have modified the text in line with this comment. We have changed the TB *LysS* to LysRS when the protein is mentioned throughout the document. We have also changed the HsKRS to KARS1 as suggested by this reviewer and Reviewer2 to provide simple differentiation between the human and TB proteins

Reviewer #4 (Remarks to the Author):

Well-written manuscript deserving of publication. This is an area of high unmet need. I agree with the target selection: the most successful drugs in TB have been protein synthesis inhibitors and drug candidates selected from this series have a higher likelihood of becoming part of a sterilising regimen. SAR is well described.

Minor comments

Why did the team choose to look for non-covalent inhibitors? A brief rationale for not looking for irreversible or suicide inhibitors would be helpful to the non-specialist reader.

The reasons we didn't pursue a covalent approach were primarily two-fold. Initially, the hit was identified as an ATP pocket binding inhibitor and there were no suitable amino acid residues, in close proximity to the compound, to target covalently. Secondly, unlike LeuRS, LysRS does not contain an editing site, to improve the fidelity of translation; therefore, that domain of the molecule cannot be targeted which is the mechanism of action of the covalent oxaborole molecule GSK3036656. We have added a sentence to highlight that the compounds bind in different sites as no editing site is present in LysRS.

The data against intra-cellular cultures of MTB is very interesting. Is there any data on intracellular concentrations?

Unfortunately, we do not have any data on intracellular concentrations

Target modification is not the only potential resistance mechanism. Resistance can also be achieved by other mechanisms: e.g., efflux pumps, metabolic bypass (how much redundancy is there in the genome?). A more thorough assessment of this is needed (a paper/in silico assessment would do)

We appreciate that there are multiple ways that resistance to this series can occur but so far, we have only seen changes in LysRS. Therefore, it is challenging to speculate about what alternative mechanisms will be involved until some are identified. We have added an extra line to the end of that section stating that the search for resistant mutants continues

Major comment

When searching for pre-existing resistance, I would want to know how globally representative the 31 clinical isolates were: for example, 31 isolates all collected from one hospital in Madrid or all from one hospital Dundee - that would be inappropriate.

Unfortunately, the lineage data is not available for these strains. These strains are all clinical isolates from Vall d'Hebron Hospital. The intention of Sup 5C was to show that "the standard mechanisms of resistance" to current clinical agents are unlikely to impact sensitivity to LysRS inhibitors; it was not to represent detailed profiling of global *M. tuberculosis* clinical strains. The legend has been modified to highlight the one source of the clinical isolates. Previously we had tested compound **7** against ~15 clinical samples from a lab in South Africa, including some with different lineages and resistance profiles, and again no impact was seen on sensitivity to **7**. To include both sets of data would have been duplicative. More in-depth clinical isolate profiling will continue as Compound **8** progresses through preclinical development.

REVIEWERS' COMMENTS

Reviewer #1 (Remarks to the Author):

In this revised version of the manuscript, the authors have addressed all the points previously raised. Therefore, I recommend publication of this improved version.

Reviewer #2 (Remarks to the Author):

The authors have fully addressed my comments to my satisfaction.